# Treatment of diabetic kidney disease. A network meta-analysis

**Fabian Büttner**[1], **Clara Vollmer Barbosa**[1], **Hannah Lang**[1], **Zhejia Tian**[1], **Anette Melk**[2‡], **Bernhard M. W. Schmidt**[1‡]*

1 Department of Nephrology and Hypertension, Hannover Medical School, Hannover, Germany,
2 Department of Pediatric Kidney, Liver and Metabolic Diseases, Hannover Medical School, Hannover, Germany

☯ These authors contributed equally to this work.
‡ AM and BMWS also contributed equally to this work.
* Schmidt.bernhard@mh-hannover.de

**Data Availability Statement:** All relevant data are within the paper and its Supporting information files.

**Funding:** BMWS received honoraria for lectures/consulting from AstraZeneca and Bayer Vital.

## Abstract

### Background

Diabetic kidney disease (DKD) is a health burden of rising importance. Slowing progression to end stage kidney disease is the main goal of drug treatment. The aim of this analysis is to compare drug treatments of DKD by means of a systemic review and a network meta-analysis.

### Methods

We searched Medline, CENTRAL and clinicaltrials.gov for randomized, controlled studies including adults with DKD treated with the following drugs of interest: single angiotensin-converting-enzyme-inhibitor or angiotensin-receptor-blocker (single ACEi/ARB), angiotensin-converting-enzyme-inhibitor and angiotensin-receptor-blocker combination (ACEi+ARB combination), aldosterone antagonists, direct renin inhibitors, non-steroidal mineralocorticoid-receptor-antagonists (nsMRA) and sodium-glucose cotransporter-2 inhibitors (SGLT2i). As primary endpoints, we defined: overall mortality and end-stage kidney disease, as secondary endpoints: renal composite outcome and albuminuria and as safety endpoints: acute kidney injury, hyperkalemia and hypotension. Under the use of a random effects model, we computed the overall effect estimates using the statistic program R4.1 and the corresponding package "netmeta". Risk of bias was assessed using the RoB 2 tool and the quality of evidence of each pairwise comparison was rated according to GRADE (Grading of Recommendations Assessment, Development and Evaluation).

### Results

Of initial 3489 publications, 38 clinical trials were found eligible, in total including 42346 patients. Concerning the primary endpoints overall mortality and end stage kidney disease, SGLT2i on top of single ACEi/ARB compared to single ACEi/ARB was the only intervention significantly reducing the odds of mortality (OR 0.81, 95%CI 0.70–0.95) and end-stage kidney disease (OR 0.69, 95%CI 0.54–0.88). The indirect comparison of nsMRA vs SGLT2i in

These funders had no role in study design, data collection and analysis, decision to publish, or preparation of the manuscript.

**Competing interests:** The authors have declared that no competing interests exist.

our composite endpoint suggests a superiority of SGLT2i (OR 0.60, 95%CI 0.47–0.76). Concerning safety endpoints, nsMRA and SGLT2i showed benefits compared to the others.

## Conclusions

As the only drug class, SGLT2i showed in our analysis beneficial effects on top of ACEi/ARB treatment regarding mortality and end stage kidney disease and by that reconfirmed its position as treatment option for diabetic kidney disease. nsMRA reduced the odds for a combined renal endpoint and did not raise any safety concerns, justifying its application.

## Introduction

Diabetes mellitus (DM) ranks among the top ten most devastating diseases worldwide, contributing to enormous disability and mortality [1]. The major burden of DM is primarily driven by its complications, among which diabetic kidney disease (DKD) is one of the most harmful. The incidence of DKD has globally substantially increased over the past three decades [2], which lead to the inevitable necessity to develop novel effective drugs to preserve kidney function, decrease the cardiovascular risk or even prevent its onset.

The treatment of DKD is based on renin-angiotensin-aldosterone system (RAAS) blockade since the 90´s. RAAS is a hormonal system that regulates the fluid and electrolyte balance and the systemic vascular system in the human body through various pathways, that affect the kidney. Drugs based on a single RAAS Blockade in particular angiotensin-converting enzyme inhibitor (ACEi) or an angiotensin receptor blocker (ARB), were the only pharmacological approach to show considerable advantages [3]. The evidence for the benefit of an additional RAAS blockade agent, either by combining ACEi+ ARB or one of them with e.g. mineralocorticoid receptor antagonists (MRA) or a direct renin inhibitor (DRI), remained limited [3]. In addition, safety concerns regarding development of hyperkalemia or acute kidney injury (abrupt changes in kidney function, including changes in serum creatinine and urine output within 48 hours or 7 days) [4] prohibited widespread use of these treatments [5].

With the emergence of a new generation of drugs (e.g. sodium-glucose cotransporter-2 inhibitors (SGLT2i), non-steroidal mineralocorticoid receptor antagonists (nsMRAs)) ground breaking impulses were given to the therapy of DKD, supported by studies showing a reduced incidence of combined renal endpoints with acceptable safety profiles [6]. SGLT2i block the carrier protein sodium-glucose linked transporter 2, which induces glucosuria and by that initiating a cascade positively affecting the cardiovascular system and kidney. nsMRA (e.g. finerenone, esaxerenone) bind highly selective on mineralocorticoid receptors and by that antagonizes the effect of aldosterone, which is steroid hormone and assumed to play a pivotal role in the development of chronic kidney disease. But, due to the lack of direct comparisons, it is still a matter of debate, which interventions show the greatest benefit in patients with DKD.

Our objective is to identify the optimal treatment strategy in addition to ACEi or ARB for patients with DKD. We mainly focused on the following questions:

- Which drug most effectively reduces the risk of overall mortality, the development of end stage kidney disease, acute kidney injury, hyperkalemia and hypotension?

- Which medication carries the least risk of inducing the specific side effects we have outlined?

To facilitate this goal, we conducted a systemic review and network meta-analysis enabling direct and indirect estimates of comparative efficacy for different interventions, specifically addressing overall mortality, incidence of end stage kidney disease (ESKD), change of albuminuria, incidence of acute kidney injury, hyperkalemia and hypotension. Thus, providing an overview of the current state of the art in the treatment of DKD and offering evidence-based guidance.

## Methods

The protocol for this analysis was registered with Prospero (CRD42021238011) and reported according to the PRISMA statements 2020 (S3 File).

We searched Cochrane Central Register of Controlled Trials (CENTRAL), Medline and clinicaltrials.gov by the following search terms: (diabetic nephropathy OR (diabetes AND kidney) OR chronic kidney disease OR CKD) AND (RCT OR randomized) combined with names of the investigated drugs (S4 File). We included only prospective, randomized, controlled clinical trials published in English from 1995 till May 2022 on with a minimum duration of 8 weeks. The deliberate choice of 8 weeks minimum allowed the inclusion of studies with specific safety endpoints (e.g., hyperkalemia). We excluded cross-over trials and retracted publications. Control groups met our inclusion criteria if they consisted of an intervention included in our analysis or a placebo on the background treatment with single ACEi/ARB.

Interventions of interest were single ACEi/ARB, ACEi+ARB combination, MRAs, DRI, nsMRAs (finerenone, esaxerenone) and SGLT2i. Due to the lack of studies fulfilling our eligibility criteria, several drug regimens (neprilysin inhibitors, high dose RAAS blockade) could not be considered. If a minimum of 80% of patients in a study were taking either ACEi or ARB, our criteria "background treatment with single ACEi/ARB" was fulfilled. Besides single ACEi/ARB, all included interventions were required to be on top of the background treatment (single ACEi or ARB treatment). All study participants were required to have diabetes mellitus and chronic kidney disease (CKD, as defined by the KDIGO criteria). Exclusion criteria were patients having other kidney diseases (e.g. glomerulonephritis), or being already on renal replacement therapy and pediatric studies.

We decided to use single ACEi/ARB as reference intervention for all comparisons, because it is the standard care for diabetic patients with CKD (KDIGO guideline 2022).

Three reviewers (FB, CV, BS) were involved in the study selection process using a predefined set of criteria through three levels of screening: title, abstract and full text. The screening at the first and second level (title & abstract) was performed by two independent authors (FB, CV) and checked by the third author (BS). At the final level ("full text") all publications were completely assessed according to our eligibility criteria and selected through consent by all reviewers (CV, BS, ZT, HL, FB, AM). No trial was excluded due to insufficient information at the abstract level. We considered as insufficient information, if a study misses for our analysis relevant outcome data, did not state if it only included patient with DKD or did not completely describe the study design (e.g. randomized, controlled). In these cases, final decisions were made at the full text level (Fig 1). Conflicts over study inclusion and data extraction were resolved by discussion with the third author (BS).

Data extraction was then performed by the two reviewers (FB, CV), subsequently the members of the team (ZT, HL, AM, BS) checked and eventually discussed the extracted data.

In case of missing data (e.g. authors did not state the background therapy or did not give enough details about study participants) study investigators were contacted via e-mail. For relevant outcome data presented only on form of figures without numerical values, we applied an image extraction software (digitizeit 2.5.; I. Bormann, Braunschweig, Germany). The following

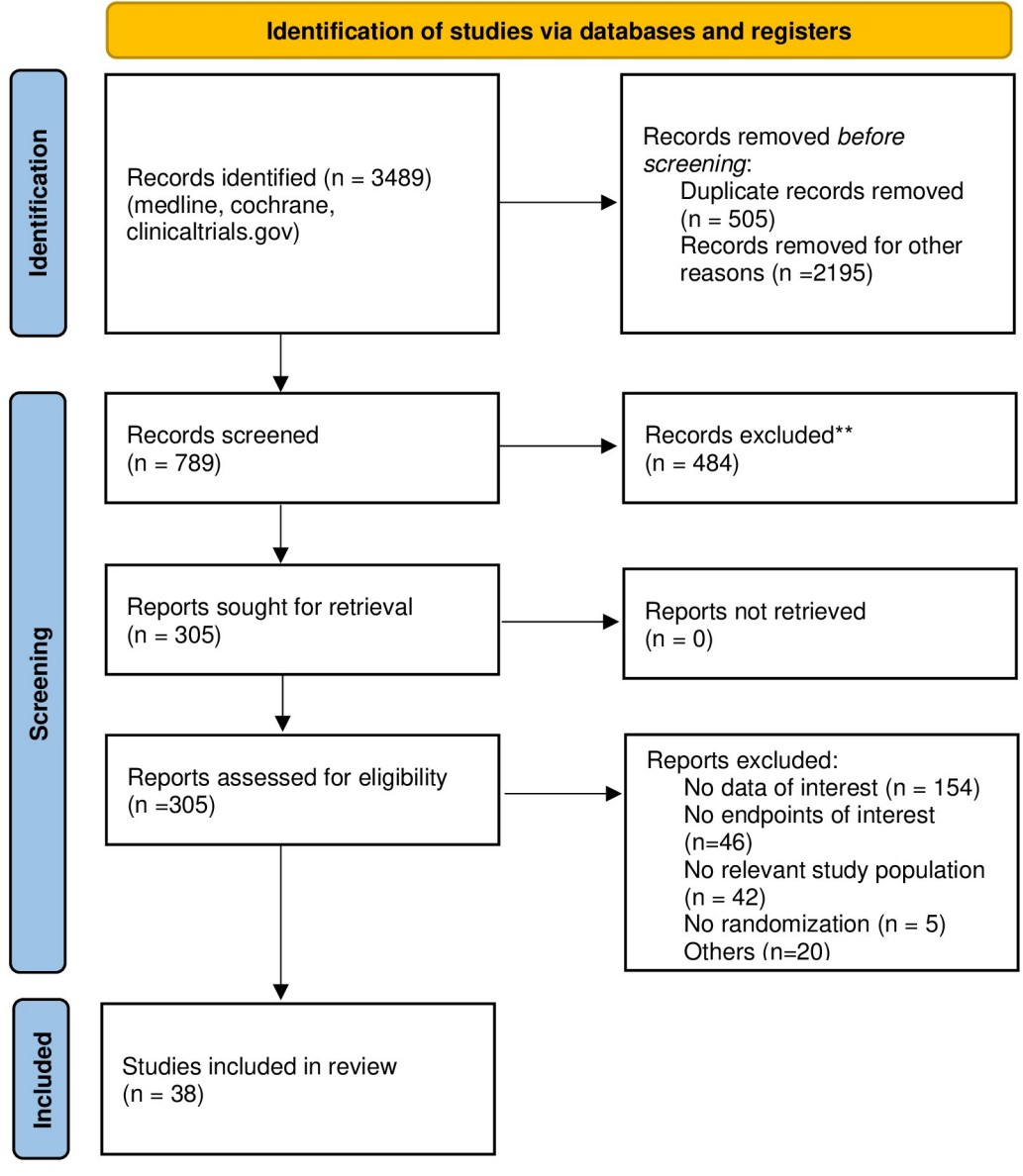

**Fig 1. Flow chart for selection process.**

characteristics were extracted study name, registration number, background therapy, type of study, inclusion criteria, exclusion criteria, study duration, study drug, dosage, mean age, percentage of male patients, outcome data and history of hypertension.

## Outcomes

Our primary outcomes were overall mortality and end stage kidney disease (ESKD). Secondary outcomes were albuminuria and a renal composite outcome (sustained eGFR of less than 15 mL/ min/1.73 m2, a sustained eGFR decline of 40% from baseline, initiation of kidney replacement therapy or kidney death). eGFR (Estimated Glomerular Filtration Rate) is a clinical measurement used in nephrology to estimate the rate at which the glomeruli are filtering waste

products and excess substances from the blood per unit of time. Safety outcomes were acute kidney injury, hyperkalemia and hypotension. Binary effect measures were estimated as odds ratio (OR) and continuous as standardized mean difference (SMD), all effect estimates are reported with 95% confidence interval (95% CI). We excluded post-hoc renal death as end-point, due to the lack of its report.

## Data analysis

Data analysis was performed with the statistic program R version 4.0.4 [7] using the "netmeta" package. Under the assumption of transitivity, we conducted a network meta-analysis using the frequentist model. A network meta-analysis is a statistical method for directly or indirectly comparing multiple interventions. It allows us to calculate the comparative effectiveness of interventions that have not been studied in head-to-head trials or have not been adequately studied. For each intervention, we applied the random effect model to generate the study effect sizes. The specific R-code and data sheet for our network meta-analysis can be found in the S15 and S16 Files.

Risk of bias was assessed by two authors using the RoB 2 tool (The Cochrane Collaboration's tool for assessing risk of bias) and additionally reporting bias through comparison adjusted funnel plots which was applied to compare older treatments with newer treatments [8, 9].

With the use of R ("netmeta" package) and Hedges´g method we computed the standardized mean difference. We computed the network plot with the "netgraph" function from the "netmeta" package, the comparison-adjusted funnel plots with the function "funnel" and the splitting of direct and indirect evidence with the "netsplit" function.

We assumed a uniformly distributed heterogeneity across the comparisons and used the $I^2$ test for the overall heterogeneity assessment, classifying the results into low heterogeneity (0–40%), moderate (40–70%) and high (70%-100%). Heterogeneity refers to the variability or differences in effect estimates across studies within a network [10]. The $I^2$ value was also computed with the "netmeta" function.

We performed a statistical analysis for design inconsistency and wherever possible for loop inconsistency. Design inconsistency stands for discrepancies in effect estimates between studies involving different composition of interventions. Loop inconsistency evaluates if direct and indirect evidence correspond with each other [10]. In order to locate hot spots of inconsistency, we performed within-design and between-design decomposition with Cochran's $Q$ statistics (significance level $\alpha = 0.05$) (S5 File) [11]. For further graphical display, we also used if suitable net heat plots. The gray squares represent the degree of importance of one treatment comparison for the estimation of another treatment comparison and the color in the background illustrates the degree of inconsistency [11]. Distribution of direct and indirect evidence was illustrated in form of tables (S6 File).

To assess the robustness of our network analysis, we conducted several sensitivity analyses [9] in order to determine how our results are affected by different input variables.

Firstly, we included only studies with low and moderate risk of bias (S10 File) to examine the influence of different levels of risk of bias on our results. Additionally, we performed a second sensitivity analysis, including only studies in which single RAAS treatment was mandatory in the control group (S11 File). This excludes studies, in which only our 80% criterion regrading ACEi/ARB background treatment was fulfilled and not all patients were receiving ACEi/ARB treatment. By changing these specific input variables, we investigated how crucial our results are impacted by our decision to include studies without a mandatory ACEi/ARB background therapy. Finally, Finerenone was examined in two major clinical trials (FIGARO,

FIDELIO). In our primary analysis we included these separately. However, as these trials had an identical design and were published also pooled in a prespecified analysis (FIDELITY), we repeated our analysis by treating the data from FIDELITY as single trial but excluding Figaro DKD and Fidelio DKD (S12 File).

Moreover, we rated the quality of evidence of each pairwise comparison according to GRADE (Grading of Recommendations Assessment, Development and Evaluation) [12]. The five following domains where implemented: Study limitations, Imprecision, Inconsistency, Indirectness and Publication bias. Every criterion for downgrading a comparison was predefined in a protocol, which may be found in the S7 File [13]. For the splitting into direct and indirect evidence we chose the back-calculation method, our subsequent results are presented in form of a forest plot for each comparison (S8 File).

### Role of the funding source

There was no funding source for this study.

## Results

Our literature search initially yielded 3489 publications, out of which 38 clinical trials were found fully eligible, in total including 42346 patients. Baseline characteristics of each study are presented in S1 File.

### Mortality

Fig 2 shows the network structure for the primary outcome overall mortality, highlighting the larger number of clinical trials for SGLT2i, nsMRAs and ACEi + ARB. As the only drug class compared to the reference intervention (single ACEi/ARB), additional SGLT2 inhibitors significantly reduced the overall mortality (OR 0.81, 95% CI 0.70–0.95; p = 0.0080) (Fig 3).

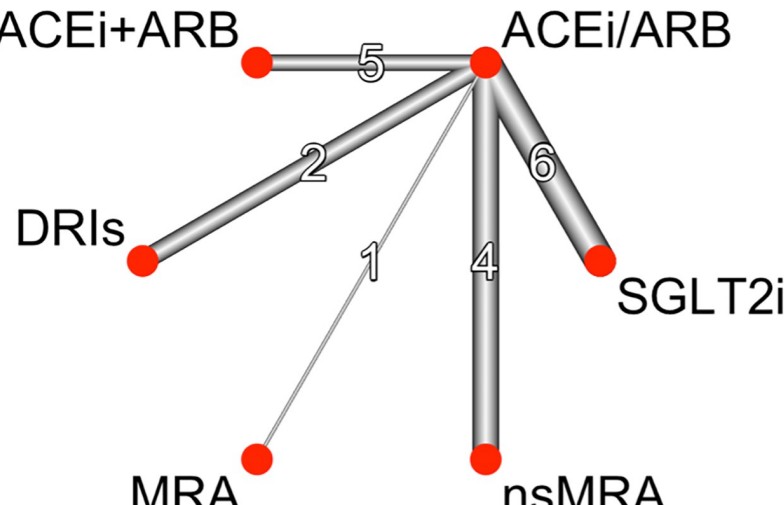

**Fig 2. Network plot for overall mortality and hyperkalemia.** The thickness of edges and numbers located on the line represent the number of trials for each direct comparison. ACEi/ARB = single Angiotensin-converting enzyme inhibitors or Angiotensin receptor blocker, ACEi+ARB = Angiotensin-converting enzyme inhibitors and Angiotensin receptor blocker combination, DRI = direct renin inhibitors, MRA = Mineralocorticoid receptor antagonists, nsMRA = non-steroidal Mineralocorticoid receptor antagonists, SGLT2i = Sodium glucose transporter inhibitors.

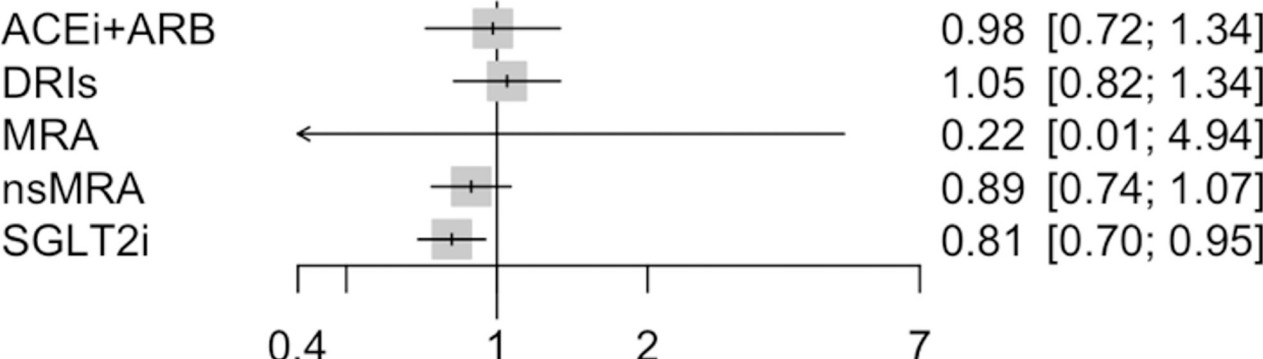

**Fig 3. Forest plot for overall mortality.** OR = Odds ratio, 95%- CI = 95% Confidence interval, single ACEi/ARB = single Angiotensin-converting enzyme inhibitors or Angiotensin receptor blocker, ACEi+ARB = Angiotensin-converting enzyme inhibitors and Angiotensin receptor blocker combination, DRI = direct renin inhibitors, MRA = Mineralocorticoid receptor antagonists, nsMRA = non-steroidal Mineralocorticoid receptor antagonists, SGLT2i = Sodium glucose transporter inhibitors.

nsMRAs (Finerenone, Esaxerenone) vs. single ACEi/ARB (OR 0.89, 95% CI 0.74–1.06) and ACEi+ARB combination (OR 0.98, 95% CI 0.72–1.34) did not show any benefits. Overall mortality was reported in only two clinical trials for DRIs (vs. reference intervention) and did not reflect any positive effects (OR 1.05, 95% CI 0.82–1.34). For MRAs (vs. reference intervention), the endpoint was only reported by one clinical trial and was rated as low quality of evidence (GRADE). Comparison of these interventions with each other did not show any significant difference (Table 1). The primary endpoint overall mortality showed a heterogeneity of $I^2 = 21.1\%$.

**Table 1. Overall mortality and end stage kidney disease.**

| ACEi/ARB | 1.22 (0.91–1.64)" | | | 1.18 (0.99–1.40)# | 1.46 (1.14–1.86)# |
|---|---|---|---|---|---|
| 1.02 (0.75–1.39)# | ACEi+ARB | | | 0.97 (0.69–1.36)* | 1.19 (0.81–1.75)* |
| 0.96 (0.75–1.22)" | 0.63 (0.63–1.39)* | DRIs | | | |
| 4.46 (0.20–97.95)* | 4.37 (0.20–97.61)§ | 4.67 (0.21–103.60)* | MRA | | |
| 1.13 (0.94–1.35)# | 1.11 (0.77–1.58)* | 1.18 (0.87–1.60)* | 0.25 (0.01–5.59)§ | nsMRA | 1.24 (0.92–1.67)* |
| 1.23 (1.06–1.43)# | 1.21 (0.86–1.70)* | 1.29 (0.97–1.72)* | 0.28 (0.01–6.10)§ | 1.09 (0.86–1.38)* | SGLT2i |

For the upper triangle (ESKD = yellow color), OR lower than 1 favors the row defining intervention (e.g. for a pairwise comparison the row defining intervention is in an alphabetical order first)

For the lower triangle (overall mortality = blue color), OR lower than 1 favors the column defining intervention

To obtain the opposing OR you may calculate the reciprocal.

GRADE (#high confidence, "moderate confidence, *low confidence, §very low confidence)

## ESKD

Eight trials reported the outcome ESKD for the drugs regimens ACEi+ARB, nsMRAs and SGLT2i. Only additional SGLT2 inhibitors on top of ACEi/ARB vs. single ACEi/ARB revealed a significant effectiveness in reducing the risk for ESKD (OR 0.69, 95% CI 0.54–0.88, p = 0.0025) (Fig 4). nsMRA did show a tendency to a reduced odds of ESKD (OR 0.85, 95% CI 0.71–1.01). Comparison of these interventions with each other did not show any significant difference (Table 1). The end stage kidney disease outcome had a heterogeneity of $I^2 = 0\%$.

## Secondary endpoints

Only MRAs on top compared to single ACEi/ARB reduced albuminuria (SMD -0.58, 95% CI -0.85, -0.31) (S13 File). For Albuminuria we computed a high heterogeneity ($I^2 = 73.9\%$).

The renal composite outcome consisting of sustained eGFR of less than 15 mL/ min/1.73 m2, sustained eGFR decline of 40% from baseline or kidney death was available for SGLT2i and nsMRAs, both reported by two trials each. In terms of improved outcomes, both interventions showed benefits (S14 File). Moreover, the indirect comparison of SGLT2i vs nsMRAs (OR 0.60, 95% CI 0.47–0.76) favors the SGLT2 inhibitors as the more effective drug class, which was rated as moderate quality of evidence (GRADE).

## Safety endpoints

We also explored the endpoint acute kidney injury (Fig 5) which was reported by a minimum of two studies for each intervention and altogether by 19 clinical trials. ACEi+ARB combination therapy demonstrated a substantially elevated risk for acute kidney injury (OR 1.69, 95% CI 1.28–2.22). Notably, SGLT2i (OR 0.93, 95% CI 0.0.77–1.12) and nsMRAs (OR 0.97; 95% CI 0.80–1.16) did not have any negative effects on renal function.

Hyperkalemia was reported by 27 trials, being one of the most frequently reported adverse events (Fig 2). In comparison to single ACEi/ARB (Fig 4), nsMRAs, MRAs and ACEi+ARB

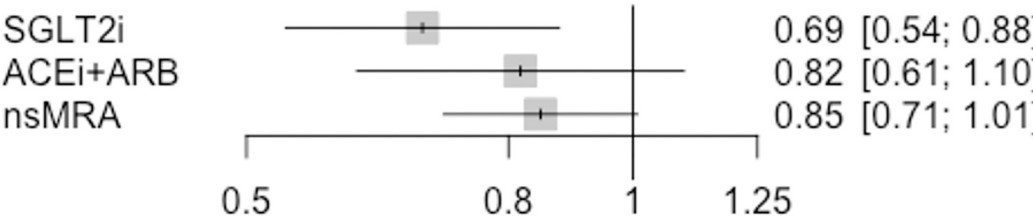

**Fig 4. Forest plot for end stage kidney disease.** OR = Odds ratio, 95%- CI = 95% Confidence interval, single ACEi/ARB = single Angiotensin-converting enzyme inhibitors or Angiotensin receptor blocker, ACEi+ARB = Angiotensin-converting enzyme inhibitors and Angiotensin receptor blocker combination, nsMRA = non-steroidal Mineralocorticoid receptor antagonists, SGLT2i = Sodium glucose transporter inhibitors.

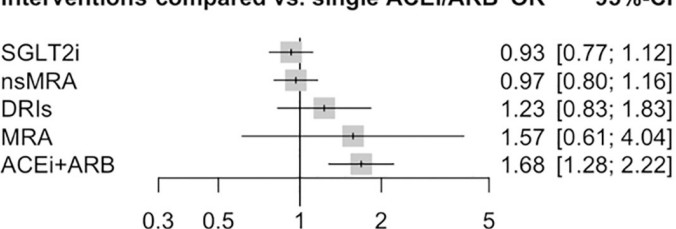

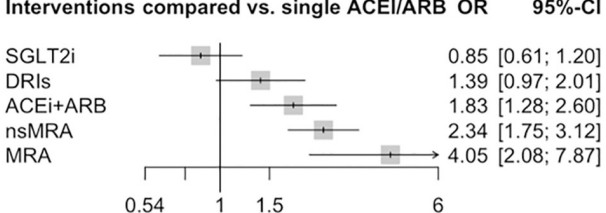

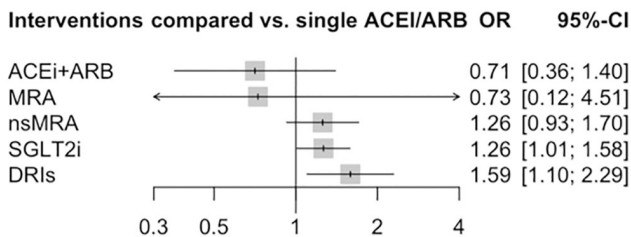

**Fig 5. Forrest plot for the following safety outcomes: Acute kidney injury, hyperkalemia, hypotension.** OR = Odds ratio, 95%- CI = 95% Confidence interval, single ACEI/ARB = single Angiotensin-converting enzyme inhibitors or Angiotensin receptor blocker, ACEi+ARB = Angiotensin-converting enzyme inhibitors and Angiotensin receptor blocker combination, DRI = direct renin inhibitors, MRA = Mineralocorticoid receptor antagonists, nsMRA = non-steroidal Mineralocorticoid receptor antagonists, SGLT2i = Sodium glucose transporter inhibitors.

combination were associated with an increased incidence of hyperkalemia. The indirect comparison of nsMRAs vs MRAs (0.58, 95% CI 0.28–1.19) reflects an attenuated risk for hyperkalemia on the side of nsMRAs, although not significant and rated as low quality of evidence (GRADE). Hyperkalemia and acute kidney injury are presented in an effect estimate table (Table 2).

SGLT2is and DRIs (Fig 5) were found to be the only two interventions to significantly elevate the risk for hypotension, (OR 1.27, 95% CI 1.01–1.59 and OR 1.59, 95% CI 1.10–2.29) respectively. In addition, nsMRAs were also associated with an increased risk (OR 1.26, 95% CI 0.92–1.71), but this finding was not significant.

**Table 2. Hyperkalemia and acute kidney injury.**

| ACEi/ARB | 0.55 (0.39–0.78)# | 0.72 (0.50–1.03)" | 0.25 (0.13–0.48)" | 0.43 (0.32–0.57)# | 1.17 (0.83–1.65)# |
|---|---|---|---|---|---|
| 0.59 (0.45–0.78)" | ACEi+ARB | 1.31 (0.79–2.20)* | 0.45 (0.23–0.91)# | 0.78 (0.50–1.23)" | 2.14 (1.31–3.49)" |
| 0.81 (0.55–1.21)" | 1.37 (0.85–2.22)* | DRIs | 0.34 (0.16–0.73)* | 0.60 (0.37–0.95)" | 1.64 (0.99–2.69)* |
| 0.64 (0.25–1.63)" | 1.07 (0.42–2.75)# | 0.78 (0.28–2.18)* | MRA | 1.73 (0.84–3.58)* | 4.75 (2.25–10.01)* |
| 1.04 (0.86–1.25)# | 1.74 (1.25–2.43)* | 1.27 (0.82–1.97)* | 1.63 (0.62–4.25)* | nsMRA | 2.74 (1.76–4.28)" |
| 1.08 (0.90–1.30)# | 1.82 (1.30–2.53)* | 1.33 (0.86–2.05)* | 1.69 (0.65–4.43)* | 1.04 (0.80–1.35)" | SGLT2i |

For the upper triangle (hyperkalemia = yellow color), OR lower than 1 favors the row defining intervention

For the lower triangle (acute kidney injury = blue color), OR lower than 1 favors the column defining intervention

To obtain the opposing OR you may calculate the reciprocal.

GRADE (#high confidence, "moderate confidence, *low confidence, §very low confidence)

## Sensitivity analyses

All three sensitivity analyses displayed a similar trend as without any exclusions (S10–S12 Files). However, the effect of nsMRA on ESKD was somewhat larger resulting in a significant result (OR 0.80, 95% CI 0.64–0.99).

Furthermore, we generally computed for all endpoints, except for albuminuria, a low heterogeneity and the Q statistics decomposition was not noticeable.

Clinical trials comprising comparisons of ACEI+ARB, MRAs and DRIs were mostly older than five years and tended to be assessed as high risk/some concerns (70%) through Cochrane´s RoB 2 tool. In contrast, SGLT2i and nsMRAs are represented mainly by recent studies with low risk of bias (appr. 85%) (S9 File). Moreover, we calculated the interrater reliability kappa on screening level (kappa = 0.88) and on abstract level (kappa = 0.89).

## Discussion

This network meta-analysis comprises data from 42346 patients with DKD. This high number allows a comprehensive analysis on the comparative effectiveness of various drug regimens. On top of treatment with ACEi or ARB only SGLT2 inhibition caused a decrease in mortality and in the incidence of ESKD. Therefore, SGLT2i should be an essential part of the treatment of each patient with DKD in addition to ACEi or ARB. We for the first time compared all interventions meant to reduce progression of DKD and have been used from nephrologists during the last two decades.

SGLT2i were introduced into treatment of DM primarily due to their glucosuric effects leading to improved glucose control [14]. Regarding DKD, this effect is only one of the potential beneficial effects of SGLT2 inhibition. In the complex pathophysiology of DKD SGLT2i exert multiple positive effects leading to reduced progression of kidney disease [15, 16]. Due to the blockade of SGLT2 less glucose and sodium is transported into the tubular cell, which in turn reduces tubular energy consumption of the tubular cell by reducing the workload to the basal Na/K ATP-ase pump and thereby reducing tubular hypoxia [17]. In addition, due to the increased amount of sodium approaching the juxtaglomerular apparatus, tubular glomerular feedback decreases renal plasma flow, glomerular pressure and glomerular filtration rate [18] leading to reduced glomerular damage. Additional positive effects of SGLT2i are their antihypertensive [19], anti-inflammatory [20] and antifibrotic properties [21]. The positive effects of SGLT2i in DKD have been comprehensively reviewed, recently [22]. Regarding safety, concerns about acute kidney injury are unfounded and SGLT2i even showed a propensity for

reducing its incidence. A meta-analysis confirmed this finding, by obtaining a significantly reduced risk for the occurrence of acute kidney injury using SGLT2i in diabetic patients [23].

Blocking the RAAS has been standard treatment of DKD. Based on the positive effects of ACEI and ARBs on DKD and overall cardiovascular risk [24] and the obvious need for further improvement of treatment, lot of effort was put in exploring interventions providing a more complete blockade of the RAAS. To provide this in addition to ACEi/ARB treatment MRAs, dual blockade with ACEi and ARB and use of ACEi/ARB in combination with aliskiren—the only available DRI—have been explored [5, 25, 26]. Unfortunately, the dual blockade with ACEi and ARB as well as the addition of aliskiren raised major safety concerns in large randomized, controlled trials [5, 26] which led to a restricted use of these two combination therapies. This is reflected by the results of our analysis, in which these treatments did not improve efficacy endpoints but showed increased safety concerns regarding acute kidney injury (ACEi + ARB) hyperkalemia (ACEi + ARB, DRIs) and hypotension (DRI).

Until recently MRAs besides Glucagon-like peptide-1 receptor agonists were the only treatment alternative for patients with progression of DKD and/or gross proteinuria despite ACEi/ARB and SGLT2i treatment. However, steroidal MRAs, i.e. spironolactone and eplerenone, were disproportionally underrepresented for sufficient data on hard endpoints in our analysis. Due to a small number of included patients and short follow up periods, most studies with MRAs were limited to albuminuria as surrogate endpoint. This suggests a positive effect on progression of DKD, but this has not been proven neither in a single study nor in our analysis. On the downside, MRAs have an unfavorable side effect profile. We can show that the odds for hyperkalemia are fourfold raised compared to ACEi/ARB monotherapy.

A further evolution of MRA, nsMRAs are the most recently developed pharmacological agents acting on the RAAS system. They block the mineralocorticoid receptor more selectively and potently than steroidal MRA (Eplerenone, Spironolactone) [27]. Due to this modified mode of action, a lower rate of hyperkalemia and a better end-organ protection were presumed [28]. Some of these preclinical assumptions were confirmed by large clinical trials [29, 30]. In our analysis, they were associated with a lower incidence of the renal composite outcome (compared to single ACEi/ARB). In comparison to the conventional single ACEi/ARB the odds for hyperkalemia were raised, but compared to steroidal MRA, they tended to show a lower incidence. Our results are not confirmative for this suggested correlation and require more studies for validation. Unlike ACEi+ARB combination or MRAs, nsMRA did not cause higher rates of acute kidney injury (compared to single ACEi/ARB). Whereas SGLT2i have already emerged as standard treatment of diabetic kidney disease [31], nsMRA are recommended in patients with persisting microalbuminuria despite sufficient ACEi/ARB treatment as additional treatment in the current consensus report of the American Diabetes Association and Kidney Disease: Improving Global Outcomes (KDIGO) [32] and the KDIGO 2022 Clinical Practice Guideline for Diabetes management in Chronic Kidney Disease [33]. First analyses of existing data are supporting this view [34, 35] and an ongoing study is exploring the combined effect of finerenone and empagliflozin [36].

Whereas albuminuria has been shown to be associated with more rapid progression of DN [37] and higher cardiovascular risk [38], interventions lowering albuminuria have not uniformly been successful in lowering the incidence of hard endpoints. For example, DRIs reduced albuminuria compared to single ACEi/ARB. The premature stop of the large clinical trial ALTITUDE, where a DRI (Aliskiren) failed to show benefits in particular for hard renal endpoints, reflects unfavorable effects of DRI [26]. Although more evidence emerges supporting albuminuria as a valid surrogate parameter for chronic kidney diseases [38], it remains disputable. Our results did not advocate use of albuminuria as a surrogate endpoint for renal outcomes as the most effective drug class with regard to renal outcome, i.e. SGLT2i, showed

the least effect on albuminuria. Notably, our result concerning albuminuria must be interpreted cautiously owing to its higher heterogeneity and raised level of inconsistency.

Limitations of our network meta-analysis were not uniformly reported endpoints like hyperkalemia, ranging from no specified cut off levels to no exact definition. Moreover, we could not consider different doses of each drug and differences in ACEi/ARB doses. In particular, most clinical trials including SGLT2i did not require the maximal labeled dose of ACEi/ARB. This however was the case in the studies including nsMRA [39]. This hampers the comparability of these drug classes. In addition, SGLT2i studies mainly had renal secondary endpoints and mostly included patients with an eGFR above 45 mL/min/1.73 m$^2$. Due to sparse data covering DRI and particularly MRA for our primary endpoints, we may offer only limited clarity about them. In general, the reporting of different renal endpoints by each clinical trial hampered the power of our NMA. For albuminuria we computed a high heterogeneity, which we assume to be mainly caused from different level of albuminuria ranging from microalbuminuria to macroalbuminuria and varying follow up periods. Several other network meta-analyses have been performed on the issue of outcome of DKD recently. However, the focus of these analyses was different from ours. Cao et al. compared SGLT2i, GLP-1RAs and dipeptidyl peptidase 4 inhibitors [40] and Ghosal et al. SGLT2i, Glucagon-like peptide– 1 receptor antagonists and finerenone [41] regarding renal endpoints. None of these focused on classical anti-proteinuric agents.

In conclusion, SGLT2i in our analysis was the only drug class showing beneficial effects on top of ACEi/ARB treatment regarding mortality and end stage kidney disease. nsMRA were based on less data but diminished the risk for our renal composite outcome without safety concerns. Our data support the actual ADA and KDIGO guidelines, which suggest that all three drug classes (ACEi/ARB, SGLT2i and nsMRA) are needed in diabetic patients with albuminuria.

## Supporting information

**S1 File. Baseline characteristics.**
(PDF)

**S2 File. Citation list.**
(PDF)

**S3 File. Prisma statements 2020.**
(PDF)

**S4 File. Search strategy.**
(PDF)

**S5 File. Tests of heterogeneity (within designs) and inconsistency (between designs) with Cochran's _Q_ statistics.**
(PDF)

**S6 File. Distribution of direct and indirect evidence.**
(PDF)

**S7 File. GRADE protocol.**
(PDF)

**S8 File. Splitting into direct and indirect evidence.**
(PDF)

**S9 File. Risk of bias (given in excel).**
(PDF)

**S10 File. Sensitivity analysis (RoB).**
(PDF)

**S11 File. Sensitivity analysis (single ACEi/ARB mandatory in control groups).**
(PDF)

**S12 File. Sensitivity analysis (replacing FIGARO and FIDELIO bei FIDELITY).**
(PDF)

**S13 File. Albuminuria, renal composite outcome (sustained eGFR <15 mL/ min/1.73 m2, sustained eGFR decline of 40% from baseline or kidney death).**
(PDF)

**S14 File. Renal composite outcome (sustained eGFR <15 mL/ min/1.73 m2, sustained eGFR decline of 40% from baseline or kidney death).**
(PDF)

**S15 File. R-code.**
(PDF)

**S16 File. Data sheets.**
(PDF)

**S17 File. GRADE for renal composite outcome.**
(PDF)

**S18 File. GRADE for hypotension.**
(PDF)

## Author Contributions

**Conceptualization:** Fabian Büttner, Clara Vollmer Barbosa, Hannah Lang, Zhejia Tian, Anette Melk, Bernhard M. W. Schmidt.

**Data curation:** Fabian Büttner, Clara Vollmer Barbosa.

**Formal analysis:** Fabian Büttner, Clara Vollmer Barbosa, Anette Melk, Bernhard M. W. Schmidt.

**Funding acquisition:** Fabian Büttner.

**Investigation:** Fabian Büttner, Clara Vollmer Barbosa, Zhejia Tian, Anette Melk, Bernhard M. W. Schmidt.

**Methodology:** Fabian Büttner, Clara Vollmer Barbosa, Hannah Lang, Anette Melk, Bernhard M. W. Schmidt.

**Project administration:** Fabian Büttner, Anette Melk, Bernhard M. W. Schmidt.

**Resources:** Fabian Büttner, Clara Vollmer Barbosa, Anette Melk, Bernhard M. W. Schmidt.

**Software:** Fabian Büttner, Clara Vollmer Barbosa.

**Supervision:** Clara Vollmer Barbosa, Anette Melk, Bernhard M. W. Schmidt.

**Validation:** Fabian Büttner, Clara Vollmer Barbosa, Bernhard M. W. Schmidt.

**Visualization:** Fabian Büttner.

**Writing – original draft:** Fabian Büttner.

**Writing – review & editing:** Fabian Büttner, Clara Vollmer Barbosa, Hannah Lang, Zhejia Tian, Anette Melk, Bernhard M. W. Schmidt.

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
