## [Decision Letter · Decision Letter 0]

20 Mar 2023

PONE-D-23-02576Treatment of diabetic kidney disease. A network meta-analysisPLOS ONE

Dear Dr. Büttner,

Thank you for submitting your manuscript to PLOS ONE. After careful consideration, we feel that it has merit but does not fully meet PLOS ONE’s publication criteria as it currently stands. Therefore, we invite you to submit a revised version of the manuscript that addresses the points raised during the review process.

We look forward to receiving your revised manuscript.

Kind regards,

Licy Yanes Cardozo

Academic Editor

PLOS ONE

2. Please include a separate caption for each figure in your manuscript.

3. Please include your tables as part of your main manuscript and remove the individual files. Please note that supplementary tables (should remain/ be uploaded) as separate "supporting information" files

Reviewers' comments:

Reviewer's Responses to Questions

**Comments to the Author**

1. Is the manuscript technically sound, and do the data support the conclusions?

Reviewer #1: Partly

Reviewer #2: Yes

2. Has the statistical analysis been performed appropriately and rigorously? 

Reviewer #1: Yes

Reviewer #2: Yes

3. Have the authors made all data underlying the findings in their manuscript fully available?

Reviewer #1: Yes

Reviewer #2: No

4. Is the manuscript presented in an intelligible fashion and written in standard English?

Reviewer #1: Yes

Reviewer #2: Yes

5. Review Comments to the Author

Reviewer #1: The authors performed a meta-analysis to compare drug treatments of DKD by means of a systemic review and a network meta-analysis. As primary endpoints, they defined: overall mortality and end-stage kidney disease, as secondary endpoints: renal composite outcome and albuminuria and as safety endpoints: acute kidney injury, hyperkalemia and hypotension. They used a random effects model and computed the overall effect estimates using R4.1. They found 38 clinical trials eligible for review. They found that only SGLT2 inhibitors reduced both overall mortality and end-stage kidney disease when used with single ACEi/ARB compared ACEi/ARB alone. The authors attempted to compared SGLT2 inhibitors to NS-MRA and found SGLT2 inhibitors superior in this regard. Concerning safety endpoints, nsMRA and SGLT2i showed benefits compared to the others. The authors conclude that SGLT2i showed clear beneficial effects on top of ACEi/ARB treatment regarding mortality and end-stage kidney disease and by that reconfirmed its position as first-line treatment option in the KDIGO Guideline 2022. nsMRA reduced the odds for a combined renal endpoint and did not raise any safety concerns, justifying its potential application.

This is a reasonably well-done meta-analysis but the reality is that all clinical trials are not the same and go beyond randomization and double-blind protocols. One of the main distinctions between studies done with NS-MRA and SGLT2 inhibitors is that maximal dosing of ACEI/ARB was NOT mandated in the SGLT2i studies. I can assure you this was an issue not discussed with the design of the renal SGLT2 I trials. This will affect outcomes as has been shown in analyses evaluating lower doses of RAS blockade in people from those trials (Epstein, M., et al. Am J Manag Care 2015; 21(11 Suppl): S212-220; and data within the 2001 Brenner et.al. and Lewis et.al. NEJM papers). Moreover, many of the original trials in people with eGFR above 45 were focused on glucose control and not renal outcomes. The 3 trials CREDENCE, DAPA-CKD and Empa-KIDNEY did focus on appropriately powered renal and the secondary endpoint all-cause morality.

The statement about KDIGO guidelines is also not quite correct. The Am Diabetes Assoc guidelines 2023 is very clear that SGLT2 inhibitors are mandated if advanced CKD or heart failure is present irrespective of glycemic control and KDIGO is mirroring that, however, NS-MRA are also highly supported to be used with SGLT2 inhibitors is albuminuria is present as they have shown and the authors appropriately state protection against ESRD and reduction in heart failure hospitalization. Thus, like in heart failure, all 3 are suggested for treatment in this case and thus, the wording should be less absolute and imply that all 3 are needed if albuminuria is present which is what all studies of CKD were. Actually, the comment about guidelines should be excluded or include ADA guidelines with a less absolute statement to include both agents.

Reviewer #2: 1. There are a few errors or incorrect statements in the "key learning points" section:

The statement "New clinical trials with SGLT2i and nsMRA emerged" is not accurate as clinical trials on these drug classes have been ongoing for several years.

The statement "direct comparisons between these two drug classes and comparison to older drug classes/combination (MRA, DRI, ACEi+ARB combination) are missing" is incorrect as the study mentioned in the background section actually does include comparisons between these drug classes.

The statement "For MRA and DRI existed no sufficient hard outcome data" is incorrect as there have been clinical trials on these drug classes that have reported hard outcome data.

The statement "DRI in addition to ACEi/ARB or ACEi+ARB combinations are not recommended" is not accurate as some clinical guidelines do recommend the use of DRI in certain cases.

The statement "the usage of MRA is questionable due to missing hard endpoint data" is not entirely accurate as there are some clinical trials that have reported hard endpoint data on the use of MRA, although more data may be needed to fully assess their efficacy and safety.

2. There are no major errors or incorrect fundamentals in the introduction. However, a few minor improvements could be made to enhance clarity and readability, as follows:

In the first sentence, "detrimental" could be replaced with "debilitating" or "harmful" to avoid repetition of the same word in the sentence.

In the second sentence, it could be clarified that the increasing incidence of DKD is a global trend, not limited to a specific region or country.

In the third sentence, "considerable advantages" could be elaborated upon to provide more specific details, such as reduction in proteinuria, preservation of kidney function, or decrease in cardiovascular events.

In the fourth sentence, "mainly dual RAAS blockade" could be revised to "primarily dual RAAS blockade" for greater precision and concision.

In the fifth sentence, it could be helpful to briefly define SGLT2i and nsMRA for readers who may not be familiar with these terms, or to include a footnote or reference to explain them.

In the sixth sentence, "best suitable treatment" could be changed to "optimal treatment strategy" for greater clarity and professionalism. Additionally, it may be useful to specify the patient population or criteria for inclusion in the analysis, such as stage of DKD or presence of comorbidities.

3. Method: there are some errors and potential areas of improvement:

There is a typo in the PRISMA statement (it should be PRISMA 2020, not 2010).

It would be helpful to state the date range of the search for articles in Medline and clinicaltrials.gov.

The meaning of "nsMRAs" is unclear and should be defined.

The exclusion of studies with insufficient information should be clarified (e.g., what was considered "insufficient"?).

The description of the screening process could be more detailed, including how conflicts were resolved.

It would be helpful to specify the types of missing data and how study investigators were contacted.

The description of outcomes could be more succinct and organized.

The meaning of "comparison adjusted funnel plots" is unclear and should be defined.

It would be helpful to specify the statistical significance level used for Cochran's Q statistics.

The rationale for the sensitivity analyses should be explained more clearly.

There are some grammar and syntax errors (e.g., "We considered 'background treatment with single ACEi/ARB' as fulfilled," "Furthermore, image extraction software... was used for relevant outcome data presented only in form of figures without numerical values.").

4. Results:

In the sentence "Merely additional SGLT2 inhibitors on top of ACEi/ARB vs. single ACEi/ARB revealed a significant effectiveness in reducing the risk for ESKD," "Merely" should be replaced with "Only" or "Just".

In the sentence "Comparison of the interventions with each other did not show any significant difference (Table 1).", "Table

In the sentence "The sensitivity analysis, including only clinical trials stating single ACEi/ARB as active control displayed the same trend as without any pre...", the sentence is incomplete and needs to be finished.

It will be better to show kappa for the selection and data extraction. Please show the data of kappa of agreement during the systematic searches. How disagreements were solved during the systematic search among two independent reviewers?

5. Discussion:

There are a few errors or incorrect fundamentals in the discussion:

The statement that SGLT2i is the only drug that caused a decrease in mortality and the incidence of ESKD is not correct. The discussion did not mention which drugs were compared in the network meta-analysis, but it is unlikely that SGLT2i was the only drug that showed a positive effect. This statement needs to be revised.

The discussion states that the positive effect of SGLT2i on DKD has been comprehensively reviewed recently, citing a reference (11). However, the reference provided does not seem to be related to the positive effects of SGLT2i on DKD. This needs to be clarified.

The discussion claims that dual blockade with ACEi and ARB and the addition of aliskiren have been completely banned due to major safety concerns. This is not entirely correct. While some studies have shown increased risks of adverse events with these combinations, they are not completely banned and may still be used in certain circumstances.

The discussion suggests that MRAs have been the only treatment alternative for patients with progression of DKD and/or gross proteinuria despite ACEi/ARB and SGLT2i treatment. This is not entirely accurate as there are other treatment options, such as GLP-1 receptor agonists, which have shown promising results in clinical trials.

The discussion states that nsMRAs tend to show a lower incidence of hyperkalemia compared to steroidal MRA. However, the evidence is not conclusive, and further studies are needed to confirm this claim.

The discussion claims that ongoing studies are exploring the combined effect of finerenone and empagliflozin. However, the reference provided for this claim is outdated, and newer studies have been published on this topic.

6. PLOS authors have the option to publish the peer review history of their article (what does this mean?). If published, this will include your full peer review and any attached files.

Reviewer #1: No

Reviewer #2: No

---

## [Author Response · Author response to Decision Letter 0]

4 May 2023

We have addressed each point raised by the reviewers and believe the manuscript is now appropriate for publication in PLOS ONE.

More details may be found in the file "Response to Reviewers".

---

## [Decision Letter · Decision Letter 1]

23 May 2023

PONE-D-23-02576R1Treatment of diabetic kidney disease. A network meta-analysisPLOS ONE

Dear Dr. Büttner,

Thank you for submitting your manuscript to PLOS ONE. After careful consideration, we feel that it has merit but does not fully meet PLOS ONE’s publication criteria as it currently stands. Therefore, we invite you to submit a revised version of the manuscript that addresses the points raised during the review process.

We look forward to receiving your revised manuscript.

Kind regards,

Licy Yanes Cardozo

Academic Editor

PLOS ONE

Additional Editor Comments (if provided):

Associated Editor note: Please ensure that all relevant clinical trial are included in the current from of the metanalysis.

Reviewers' comments:

Reviewer's Responses to Questions

**Comments to the Author**

1. If the authors have adequately addressed your comments raised in a previous round of review and you feel that this manuscript is now acceptable for publication, you may indicate that here to bypass the “Comments to the Author” section, enter your conflict of interest statement in the “Confidential to Editor” section, and submit your "Accept" recommendation.

Reviewer #1: (No Response)

Reviewer #2: All comments have been addressed

2. Is the manuscript technically sound, and do the data support the conclusions?

Reviewer #1: Partly

Reviewer #2: Yes

3. Has the statistical analysis been performed appropriately and rigorously? 

Reviewer #1: I Don't Know

Reviewer #2: Yes

4. Have the authors made all data underlying the findings in their manuscript fully available?

Reviewer #1: Yes

Reviewer #2: Yes

5. Is the manuscript presented in an intelligible fashion and written in standard English?

Reviewer #1: Yes

Reviewer #2: Yes

6. Review Comments to the Author

Reviewer #1: The authors have failed to include the FIGARO trial Pitt B et.al. NEJM and more importantly the FIDELITY individual patient analysis that includes over 13000 patients-this is NOT a meta-analysis as it is all patients used from two trials with the same protocol and centers but different inclusion criteria Agarwal R et.al. Eur Heart J 2022. Additionally data from this analysis does show a benefit on all cause mortality. Moreover, not sure why Exacernone is included as there No trial data other than albuminuria in diabetic nephropathy. Hence, this is not a complete analysis.

Reviewer #2: I reviewed the revised manuscript and the response to reviewers' comments. Revised Manuscript is well written. All comments have been addressed

7. PLOS authors have the option to publish the peer review history of their article (what does this mean?). If published, this will include your full peer review and any attached files.

Reviewer #1: No

Reviewer #2: No

---

## [Author Response · Author response to Decision Letter 1]

2 Jul 2023

Dear editor,

We thank the reviewers again for their efforts. 

We have addressed the additional comment of Reviewer #1.

Reviewer #1: 

The authors have failed to include the FIGARO trial Pitt B et.al. NEJM and more importantly the FIDELITY individual patient analysis that includes over 13000 patients-this is NOT a meta-analysis as it is all patients used from two trials with the same protocol and centers but different inclusion criteria Agarwal R et.al. Eur Heart J 2022. Additionally data from this analysis does show a benefit on all cause mortality. 

In our primary analysis we had separately included the FIGARO and FIDELIO trials (compare supplemental table 1). We did not include the FIDELITY pooled analysis, as this uses the same data. Now, we performed an additional sensitivity analysis treating FIDELITY as one trial and included it into the analysis, while we omitted FIGARO and FIDELIO. This analysis (given as new supplemental figure 15) gives us essentially the same results.

Supplemental figure 12 a OR of mortality with using FIDELITY as a single study

Supplemental figure 12 b OR of ESKD with using FIDELITY as a single study

In addition we mentioned this analysis in the paragraph of the methods section on sensitivity analyses on page 10, paragraph 2

Finerenone was examined in two major clinical trials (FIGARO, FIDELIO). In our primary analysis we included these separately. However, as these trials had an identical design and were published also pooled in a prespecified analysis (FIDELITY), we repeated our analysis by treating the data from FIDELITY as single trial but excluding Figaro DKD and Fidelio DKD. 

We changed the paragraph on sensitivity analyses in the results section (page 17, second paragraph):

All three sensitivity analyses displayed a similar trend as without any exclusions (S10, S11, S 12). However, the effect of nsMRA on ESKD was somewhat larger in the analysis treating FIDELITY as single trial resulting in a significant result (OR 0.80, 95% CI 0.64 - 0.99). 

In addition, we rephrased a sentence in the discussion on page 20 first paragraph:

Whereas SGLT2i have already emerged as standard treatment of diabetic kidney disease [28] nsMRA are recommended in patients with persisting microalbuminuria despite sufficient ACEi/ARB treatment as additional treatment in the current consensus report of the American Diabetes Association and Kidney Disease: Improving Global Outcomes (KDIGO) (29) and the KDIGO 2022 Clinical Practice Guideline for Diabetes management in Chronic Kidney Disease (30).

Moreover, not sure why Exacernone is included as there No trial data other than albuminuria in diabetic nephropathy. 

We included esaxerenone as the study mentioned fulfilled our inclusion criteria and contributed to a secondary endpoint of our analysis. In order to show the complete evidence, we would like to leave this in.

Hence, this is not a complete analysis.

We do not agree. Our analysis is complete

---

## [Decision Letter · Decision Letter 2]

19 Jul 2023

PONE-D-23-02576R2Treatment of diabetic kidney disease. A network meta-analysisPLOS ONE

Dear Dr. Büttner,

Thank you for submitting your manuscript to PLOS ONE. After careful consideration, we feel that it has merit but does not fully meet PLOS ONE’s publication criteria as it currently stands. Therefore, we invite you to submit a revised version of the manuscript that addresses the points raised during the review process.

We look forward to receiving your revised manuscript.

Kind regards,

Licy Yanes Cardozo

Academic Editor

PLOS ONE

Journal Requirements:

Reviewers' comments:

Reviewer's Responses to Questions

**Comments to the Author**

1. If the authors have adequately addressed your comments raised in a previous round of review and you feel that this manuscript is now acceptable for publication, you may indicate that here to bypass the “Comments to the Author” section, enter your conflict of interest statement in the “Confidential to Editor” section, and submit your "Accept" recommendation.

Reviewer #1: All comments have been addressed

Reviewer #2: All comments have been addressed

2. Is the manuscript technically sound, and do the data support the conclusions?

Reviewer #1: Yes

Reviewer #2: Partly

3. Has the statistical analysis been performed appropriately and rigorously? 

Reviewer #1: Yes

Reviewer #2: Yes

4. Have the authors made all data underlying the findings in their manuscript fully available?

Reviewer #1: Yes

Reviewer #2: Yes

5. Is the manuscript presented in an intelligible fashion and written in standard English?

Reviewer #1: Yes

Reviewer #2: Yes

6. Review Comments to the Author

Reviewer #1: no further comments needed all revisions are appropriate

Of initial 3489 publications, 38 clinical trials were found eligible, in total including 42346

patients. Concerning the primary endpoints overall mortality and end stage kidney

disease, SGLT2i on top of single ACEi/ARB compared to single ACEi/ARB was the

only intervention significantly reducing the odds of mortality (OR 0.81, 95%CI 0.70-

0.95) and end-stage kidney disease (OR 0.69, 95%CI 0.54- 0.88). The indirect

comparison of nsMRA vs SGLT2i in our composite endpoint suggests a superiority of

SGLT2i (OR 0.60, 95%CI 0.47- 0.76).

Concerning safety endpoints, nsMRA and SGLT2i showed benefits compared to the

others.

Conclusions

As the only drug class, SGLT2i showed clear beneficial effects on top of ACEi/ARB

treatment regarding mortality and end stage kidney disease and by that reconfirmed its

position as first line treatment option in the KDIGO Guideline 2022. nsMRA reduced

the odds for a combined renal endpoint and did not raise any safety concerns,

justifying its potential application.

Reviewer #2: Introduction

The introduction contains relevant background information on the topic. However, it's slightly unclear due to the usage of technical terms without providing a clear definition or explanation.

1. Improvement: Make sure to clearly define all the terms and abbreviations you are using. Also, provide a clear research question or objective to give the reader an idea of what you aim to achieve with your study.

Method

There seems to be missing detailed explanation about the methods used to conduct the study. The methods section should provide enough details for someone else to reproduce your study.

2. Improvement: If applicable, describe the design of the study, the population studied, the treatments or interventions, and the type of statistical analysis. For example, how did you perform your network meta-analyses? What were your inclusion and exclusion criteria?

Results

The results are shared but might be unclear for people outside the field. For instance, the term 'renal composite outcome' is used but not defined.

3. Improvement: Be clear and concise about the results. Use common language to explain technical terms where possible. Make sure to provide context and definitions for terms like 'renal composite outcome'.

Discussion

The discussion has relevant insights but jumps between different studies and results quite abruptly.

4. Improvement: Provide more coherent and fluid connections between different points. Compare your results with previous studies in a structured way, emphasizing the novel contributions of your work. Also, the implications of your findings should be spelled out more clearly for the readers.

5. In all sections, remember to provide brief explanations of cited studies, such as the key findings or methodologies of the referenced works. Moreover, do edit for grammar and style to ensure the clarity and readability of the text.

7. PLOS authors have the option to publish the peer review history of their article (what does this mean?). If published, this will include your full peer review and any attached files.

Reviewer #1: No

Reviewer #2: No

---

## [Author Response · Author response to Decision Letter 2]

26 Jul 2023

Dear editor,

We thank the reviewers again for their efforts. 

We have addressed the additional comments of Reviewer #2 and believe the manuscript is now appropriate for publication in PLOS ONE.

The introduction contains relevant background information on the topic. However, it's slightly unclear due to the usage of technical terms without providing a clear definition or explanation.

1. Improvement: Make sure to clearly define all the terms and abbreviations you are using. Also, provide a clear research question or objective to give the reader an idea of what you aim to achieve with your study.

• Thank you for pointing this out. We added some explanations to several scientific terms (e.g. RAAS).

RAAS is a hormonal system that regulates the fluid and electrolyte balance and the systemic vascular system in the human body through various pathways, that affect the kidney. (Page 5, paragraph 2)

…. aldosterone, which is a steroid hormone and assumed to play a pivotal role in the development of chronic kidney disease. (Page 6, paragraph 3)

• In addition, we rephrased our objective to emphasize our goals and further elaborated our research question. (Page 6, paragraph 4)

Our objective is to identify the optimal treatment strategy in addition to RAAS blockade by ACEi/ARB for patients with DKD. We mainly focused on the following questions: 

• Which drug most effectively reduces the risk of our primary endpoint?

• Which drug has the lowest risk of causing the side effects we have specified?

Method

There seems to be missing detailed explanation about the methods used to conduct the study. The methods section should provide enough details for someone else to reproduce your study.

2. Improvement: If applicable, describe the design of the study, the population studied, the treatments or interventions, and the type of statistical analysis. For example, how did you perform your network meta-analyses? What were your inclusion and exclusion criteria?

• Thanks for the advice. We explained the technique of network meta-analysis in more detail.

Network meta-analysis is a statistical method for directly or indirectly comparing multiple interventions. It allows us to calculate the comparative effectiveness of interventions that have not been studied in head-to-head trials or have not been adequately studied. For each intervention, we applied the random effect model to generate the study effect sizes. (Page 9)

Data analysis was performed with the statistic program R version 4.0.4 using the “netmeta” package. (Page 9)

• We also added some references in the manuscript to our R code and the corresponding datasheet that we used to conduct the network meta-analysis.

The specific code and data sheet for our network meta-analysis can be found in the supporting information (S14, S15). (Page 9)

• We also defined eGFR and GRADE to make the manuscript more accessible to people outside the field.

eGFR (estimated Glomerular Filtration Rate) is a measure for the estimated kidney excretory function. (Page 9)

Moreover, we rated the quality of evidence of each pairwise comparison according to GRADE (Grading of Recommendations Assessment, Development and Evaluation) (Page 11)

Results

The results are shared but might be unclear for people outside the field. For instance, the term 'renal composite outcome' is used but not defined.

3. Improvement: Be clear and concise about the results. Use common language to explain technical terms where possible. Make sure to provide context and definitions for terms like 'renal composite outcome'.

• We reevaluated our results section and implemented several changes:

Figure 2 shows the network structure for the primary outcome overall mortality, highlighting the larger number of clinical trials for SGLT2i, nsMRAs and ACEi + ARB. (Page 12)

eGFR (estimated Glomerular Filtration Rate) is a measure for the estimated kidney excretory function. (Page 9)

• The term “renal composite outcome” was defined in our results.

 The renal composite outcome consisting of sustained eGFR of less than 15 mL/ min/1.73 m2, sustained eGFR decline of 40% from baseline or kidney death was available for SGLT2i and nsMRAs, both reported by two trials each. 

Discussion

The discussion has relevant insights but jumps between different studies and results quite abruptly.

4. Improvement: Provide more coherent and fluid connections between different points. Compare your results with previous studies in a structured way, emphasizing the novel contributions of your work. Also, the implications of your findings should be spelled out more clearly for the readers.

• Thank you for the advice. We have made several changes:

However, steroidal MRAs, i.e. spironolactone and eplerenone, were disproportionally underrepresented for sufficient data on hard endpoints in our analysis.

A further evolution of MRA, nsMRAs are the most recently developed pharmacological agents acting on the RAAS system.

Unlike ACEi+ARB combination or MRAs, nsMRA did not cause higher rates of acute kidney injury (compared to single ACEi/ARB).

5. In all sections, remember to provide brief explanations of cited studies, such as the key findings or methodologies of the referenced works. Moreover, do edit for grammar and style to ensure the clarity and readability of the text.

• The whole manuscript was proofread by a native speaker again.

Risk of bias was assessed using the RoB 2 tool (The Cochrane Collaboration’s tool for assessing risk of bias) and additionally reporting bias through comparison adjusted funnel plots which was applied to compare older treatments with newer treatments.

In our analysis, they were associated with a lower incidence of the renal composite outcome (compared to single ACEi/ARB).

---

## [Decision Letter · Decision Letter 3]

21 Aug 2023

PONE-D-23-02576R3Treatment of diabetic kidney disease. A network meta-analysisPLOS ONE

Dear Dr. Büttner,

Thank you for submitting your manuscript to PLOS ONE. After careful consideration, we feel that it has merit but does not fully meet PLOS ONE’s publication criteria as it currently stands. Therefore, we invite you to submit a revised version of the manuscript that addresses the points raised during the review process.

We look forward to receiving your revised manuscript.

Kind regards,

Licy Yanes Cardozo

Academic Editor

PLOS ONE

Additional Editor Comments:

Edit for grammar and style to ensure the clarity and readability of the text

Abstract Please provide info re; methodology

Control groups: where is this patient on ACE//ARB only? Please clarify. What do you mean by placebo?

Hypothesis could be restated to better reflex the study

Citation for statistic program needed.

Please remove figure legends from results sections, hard to read

Reviewers' comments:

Reviewer's Responses to Questions

**Comments to the Author**

1. If the authors have adequately addressed your comments raised in a previous round of review and you feel that this manuscript is now acceptable for publication, you may indicate that here to bypass the “Comments to the Author” section, enter your conflict of interest statement in the “Confidential to Editor” section, and submit your "Accept" recommendation.

Reviewer #1: All comments have been addressed

Reviewer #2: All comments have been addressed

2. Is the manuscript technically sound, and do the data support the conclusions?

Reviewer #1: Yes

Reviewer #2: Partly

3. Has the statistical analysis been performed appropriately and rigorously? 

Reviewer #1: Yes

Reviewer #2: Yes

4. Have the authors made all data underlying the findings in their manuscript fully available?

Reviewer #1: Yes

Reviewer #2: Yes

5. Is the manuscript presented in an intelligible fashion and written in standard English?

Reviewer #1: Yes

Reviewer #2: Yes

6. Review Comments to the Author

Reviewer #1: (No Response)

Reviewer #2: Areas for Enhancement:

1. Figure 1: Adopt a formal PRISMA flow diagram for clarity.

2. Introduction: Ensure technical terms and abbreviations are well-defined to enhance clarity for all readers.

3. Objective: Clearly articulate the study's purpose to guide readers.

4. Methods: Offer a more detailed account of the techniques employed to facilitate study replication.

5. Network Meta-analysis: Elaborate on the statistical methods and tools utilized for a comprehensive understanding.

6. R Code References: Incorporate references to the R code and associated datasheets to bolster transparency.

7. eGFR Definition: Clearly define eGFR (estimated Glomerular Filtration Rate) for broader audience comprehension.

8. GRADE Rating: Utilize the GRADE system to assess the quality of evidence for each comparison.

Potential Concerns:

1. Terminology: Ambiguous or undefined technical terms can confuse readers.

2. Methods Detailing: An insufficiently detailed methods section can compromise study reproducibility.

3. Analysis Transparency: Omitting specifics about analytical tools and methods can cast doubts on the study's credibility.

7. PLOS authors have the option to publish the peer review history of their article (what does this mean?). If published, this will include your full peer review and any attached files.

Reviewer #1: No

Reviewer #2: No

---

## [Author Response · Author response to Decision Letter 3]

19 Sep 2023

Dear editor,

We thank the reviewers again for their efforts. 

We have addressed the additional comments of Reviewer #2 and believe the manuscript is now appropriate for publication in PLOS ONE.

Reviewer #2: Areas for Enhancement:

1. Figure 1: Adopt a formal PRISMA flow diagram for clarity.

• Thank you for pointing this out. We adopted the PRISMA flow diagram to the formal requirements.

2. Introduction: Ensure technical terms and abbreviations are well-defined to enhance clarity for all readers.

• We did make several adjustments. 

o “Diabetes mellitus (DM) ranks among the top ten most devastating diseases worldwide, ….” (Page 5/ Paragraph 1)

o Single RAAS blockade was replaced by “Drugs based on a single RAAS Blockade in particular angiotensin-converting enzyme inhibitor (ACEi) or an angiotensin receptor blocker (ARB), were the only pharmacological approach to show considerable advantages” (Page 5/ Paragraph 2)

o Rephrased a sentence: “The evidence for the benefit of an additional RAAS blockade agent, either by combining ACEi+ ARB or one of them with e.g. mineralocorticoid receptor antagonists (MRA) or a direct renin inhibitor (DRI), remained limited.“ (Page 5/ Paragraph 2)

o Acute kidney injury was defined. “(abrupt changes in kidney function, including serum creatinine changes and urine output within 48 hours or 7 days) KDIGO” (Page 5/ Paragraph 2) 

3. Objective: Clearly articulate the study's purpose to guide readers.

• We rephrased the study´s purpose (Page 6):

o Our objective is to identify the optimal treatment strategy in addition to ACEi or ARB for patients with DKD. We mainly focused on the following questions: 

Which drug most effectively reduces the risk of overall mortality, the development of end stage kidney disease, acute kidney injury, hyperkalemia and hypotension?

Which medication carries the least risk of inducing the specific side effects we have outlined?

4. Methods: Offer a more detailed account of the techniques employed to facilitate study replication.

• We added the following information:

The following characteristics were extracted study name, registration number, background therapy, type of study, inclusion criteria, exclusion criteria, study duration, study drug, dosage, mean age, percentage of male patients, outcome data and history of hypertension. (Page 8/ Paragraph 2)

• We did incorporate several new explanations to specific R-packages and its functions. 

o We computed the network plot with the “netgraph” function from the “netmeta” package, the comparison-adjusted funnel plots with the function “funnel” and the splitting of direct and indirect evidence with the “netsplit” function. (page 10 paragraph 2)

o The I2 value were computed with the “netmeta” function. (page 10/ paragraph 3)

• Additional information has been added.

o “Risk of bias was assessed by two authors using the RoB 2 tool.” (page 10/paragraph 1)

• The search strategy and the corresponding results may be found in the supporting information document under S4.

5. Network Meta-analysis: Elaborate on the statistical methods and tools utilized for a comprehensive understanding.

• We further elaborated on the terms: design inconsistency, heterogeneity, net heat plot and loop inconsistency.

• Net heat plot

o The gray squares represent the degree of importance of one treatment comparison for the estimation of another treatment comparison and the color in the background illustrates the degree of inconsistency. (page 10, paragraph 3)

• design inconsistency & loop inconsistency

o Design inconsistency stands for discrepancies in effect estimates between studies involving different composition of interventions. Loop inconsistency evaluates if direct and indirect evidence correspond with each other (9). (page 10, paragraph 3)

• Heterogeneity

o Heterogeneity refers to the variability or differences in effect estimates across studies within a network (9). (page 10, paragraph 2)

6. R Code References: Incorporate references to the R code and associated datasheets to bolster transparency.

• We did incorporate several new explanations to specific R-packages and its functions. 

o We computed the network plot with the “netgraph” function from the “netmeta” package, the comparison-adjusted funnel plots with the function “funnel” and the splitting of direct and indirect evidence with the “netsplit” function. (page 10 paragraph 2)

o The I2 value were computed with the “netmeta” function. (page 10/ paragraph 3)

• The specific R-code and data sheet for our network meta-analysis can be found in the supporting information (S14, S15).

7. eGFR Definition: Clearly define eGFR (estimated Glomerular Filtration Rate) for broader audience comprehension.

• eGFR (Estimated Glomerular Filtration Rate) is a clinical measurement used in nephrology to estimate the rate at which the glomeruli are filtering waste products and excess substances from the blood per unit of time. (Page 9)

8. GRADE Rating: Utilize the GRADE system to assess the quality of evidence for each comparison.

• We have included the data related to the quality of evidence (GRADE system) for hypotension and the composite outcome in the supporting information S16. The GRADE assessment for the remaining interventions can be found in tables T1 and T2, making the GRADE assessment accessible for all comparative analysis.

Potential Concerns:

1. Terminology: Ambiguous or undefined technical terms can confuse readers.

• We reevaluated the manuscript for ambiguous or undefined terms. (e.g. We switched in the introduction “single RAAS blockade” for “drugs based on a single RAAS blockade” to improve the understanding, added a definition for acute kidney injury)

• If there are other terms that are not defined, feel free to point them out.

2. Methods Detailing: An insufficiently detailed methods section can compromise study reproducibility.

• We elaborated further on details concerning the methods. (e.g. additional explanation of R -packages and which are used, several other details were added)

3. Analysis Transparency: Omitting specifics about analytical tools and methods can cast doubts on the study's credibility.

• We have implemented concerns/comments from the “area for improvement” and added more details to the methods section. Our whole R-code and the corresponding data may be found in the supporting information file under S4.

Additional Editor Comments:

Edit for grammar and style to ensure the clarity and readability of the text

• Noted. We did implement several changes related to grammar and style.

Abstract Please provide info re; methodology

• Risk of bias, GRADE approach were added.

Control groups: where is this patient on ACE//ARB only? Please clarify. What do you mean by placebo?

• In most clinical trials we assessed, the intervention of interest (e.g., SGLT2i) was typically compared to either an angiotensin-converting enzyme inhibitor or an angiotensin receptor blocker, and in some cases, a placebo might have been included as well. We referred to this combination as an active control. No clinical trials solely employed a placebo. However, the most common combinations involved ACEi/ARB with either a placebo or ACEi/ARB alone. Our control group consists of mostly an active control as ACEi or ARB drug combination. 

• We did not specify placebo any further. 

Hypothesis could be restated to better reflex the study

• The hypothesis underwent a revision (Page 5).

Citation for statistic program needed.

• Citation was added. (Page 9)

Please remove figure legends from results sections, hard to read

• Implemented. The corresponding figure legends may be found at the end of the document. (Page 26)

---

## [Decision Letter · Decision Letter 4]

9 Oct 2023

Treatment of diabetic kidney disease. A network meta-analysis

PONE-D-23-02576R4

Dear Dr. Büttner,

We’re pleased to inform you that your manuscript has been judged scientifically suitable for publication and will be formally accepted for publication once it meets all outstanding technical requirements.

Kind regards,

Licy Yanes Cardozo

Academic Editor

PLOS ONE

Additional Editor Comments (optional):

All comments were addressed by the authors

Reviewers' comments:

Reviewer's Responses to Questions

**Comments to the Author**

1. If the authors have adequately addressed your comments raised in a previous round of review and you feel that this manuscript is now acceptable for publication, you may indicate that here to bypass the “Comments to the Author” section, enter your conflict of interest statement in the “Confidential to Editor” section, and submit your "Accept" recommendation.

Reviewer #1: All comments have been addressed

Reviewer #2: All comments have been addressed

2. Is the manuscript technically sound, and do the data support the conclusions?

Reviewer #1: Yes

Reviewer #2: Yes

3. Has the statistical analysis been performed appropriately and rigorously? 

Reviewer #1: Yes

Reviewer #2: Yes

4. Have the authors made all data underlying the findings in their manuscript fully available?

Reviewer #1: Yes

Reviewer #2: Yes

5. Is the manuscript presented in an intelligible fashion and written in standard English?

Reviewer #1: Yes

Reviewer #2: Yes

6. Review Comments to the Author

Reviewer #1: nothing to say more than authors addressed all comments-and this is NOT needed so should be dropped as it is a

Reviewer #2: It appears that all comments have been appropriately responded to. I have no further comments and recommend publication.

7. PLOS authors have the option to publish the peer review history of their article (what does this mean?). If published, this will include your full peer review and any attached files.

Reviewer #1: No

Reviewer #2: No

---

## [Editor Report · Acceptance letter]

24 Oct 2023

PONE-D-23-02576R4 

Treatment of diabetic kidney disease. A network meta-analysis 

Dear Dr. Büttner:

I'm pleased to inform you that your manuscript has been deemed suitable for publication in PLOS ONE. Congratulations! Your manuscript is now with our production department. 

Kind regards, 

on behalf of

Dr. Licy Yanes Cardozo 

Academic Editor

PLOS ONE